# Molecular Characterization and Genetic Variation in *Ceratocystis fimbriata* Ell. and Halst. on Pomegranate

**DOI:** 10.3390/jof8121276

**Published:** 2022-12-05

**Authors:** Navyashree Sailapura Erakyathappa, Devappa Venkatappa, Sangeetha Chittarada Gopal, Shankarappa Sridhara, Honnabyraiah Madhapura Kalaiah, Jayashree Ugalat

**Affiliations:** 1Department of Plant Pathology, College of Horticulture, Bengaluru, University of Horticultural Sciences, Bagalkot 560 065, India; 2Center for Climate Resilient Agriculture, Keladi Shivappa Nayaka University of Agricultural and Horticultural Sciences, Shivamogga 577 201, India; 3Department of Fruit Science, College of Horticulture, Mysuru, University of Horticultural Sciences, Bagalkot 571 130, India; 4Department of Biotechnology and Crop Improvement, College of Horticulture, Bengaluru, University of Horticultural Sciences, Bagalkot 560 065, India

**Keywords:** variability, characterization, *Ceratocystis fimbriata*, ITS primers

## Abstract

Fifteen isolates of *Ceratocystis fimbriata* collected from different locations in Karnataka were characterized using ITS gene technology. It produced an amplification size of 600–650 bp, which indicated that all the isolates belong to the genus *Ceratocystis*, thus confirming the identity of the pathogenic isolates. To test genetic variability, isolates were analyzed using microsatellite markers. An UPGMA dendrogram for genetic variation among the isolates showed that all the isolates fell into two major clusters. The first cluster consisted of isolate Cf-10 and the second cluster was further divided into two sub-clusters. Sub-cluster one consisted of isolate Cf-2. Sub-cluster two was again divided into five groups. The first group included isolate Cf-13, the second group consisted of isolate Cf-14, the third group included isolates Cf-1, Cf-4, Cf-6, Cf-7, Cf-8 and Cf-9, the fourth group included Cf-5 and Cf-11, and the fifth group consisted of Cf-3, Cf-12 and Cf-15. The dissimilarity coefficient ranged from 0.00 to 0.20 among the isolates. Isolates Cf-1, Cf-3, Cf-4, Cf-5 Cf-6, Cf-7, Cf-8, Cf-9, Cf-11, Cf-12 and Cf-15 were found to be highly similar, as their dissimilarity coefficient was zero. Maximum dissimilarity (0.20) was found between isolate Cf-10 and all the other isolates, suggesting they were genetically distinct.

## 1. Introduction

Pomegranate (*Punica granatum* L.), one of the oldest known fruits belonging to the family Punicaceae, has high medicinal value. The fruit is rich in antioxidants and symbolizes health, fertility and eternal life. Pomegranate is considered one of the vital cash crops of India due to its growing demand and export value around the world. India is one of the major producers of pomegranate in the world, having an annual production of 3034 thousand metric tonnes with 262 thousand hectares of cultivable area and productivity of 11.58 metric tonnes per hectare. It is commercially cultivated in Maharashtra, Karnataka, Gujarat, Andhra Pradesh, Madhya Pradesh, Rajasthan and Tamil Nadu. Karnataka is the second leading producer of the country, having an area of 25.97 thousand hectares with a production of 268.23 thousand metric tonnes and productivity of 10.32 metric tonnes. The crop has extended across different districts, including Chitradurga, Ballari, Tumakuru, Vijayapura, Bagalkote, Koppala, Belagavi, Davanagere, Bengaluru and Kalaburagi (Anon. [1]). Among the diseases infecting pomegranates, wilt caused by *Ceratocystis Fimbriata* (Cf) is a major limiting factor for pomegranate production in all the country’s growing areas. It has been said the disease will bring about the fall of the pomegranate, which indicates the dying of pomegranate trees. The disease was first reported in India in the Nashik District of Maharashtra in 1978 and later the Kaladagi and Kanamadi areas of Karnataka on the soft-seeded Ganesh variety in 1988 [2]. The *Ceratocystis* sp. was isolated from the infected stem, root and branch tissues collected from various locations during 1996. The fungus isolated from Bagalkot was confirmed by the International Mycological Institute (UK) as *C. Fimbriata* by Ellis and Halst in 1997 [3]. During the survey, fifteen isolates of *C. fimbriata* were isolated from 24 pomegranate orchards. The isolates’ identity was confirmed using 16 microsatellites (simple sequence repeats, SSR) markers and ITS primers.

It is necessary to investigate the molecular variability and identification of *C. Fimbriata* using molecular markers to understand the phylogenetic relationship among the isolates. It is also fundamental to guide the development of appropriate strategies for disease management. The objective of this study was to evaluate the genetic diversity and possible origins of *C. Fimbriata* using polymorphic microsatellite markers.

## 2. Material and Methods

### 2.1. Fungal Isolation and Molecular Characterization

The infected samples of pomegranate were collected from major pomegranate-growing regions in the southern districts of Karnataka, including rural Bengaluru, Bengaluru, Ramanagara, Chamarajanagara, Tumakuru and Chitradurga, from 2019 to 2020. The isolations were performed following the carrot bait technique [4], where tissue bits were enclosed in a cavity hollowed out of the inner face of a pair of disks cut from carrots, which had previously been washed in running tap water. The disks were fastened with a sterile rubber band and incubated at high relative humidity (RH) for 4 days at 26 ± 1 °C.

### 2.2. Koch’s Postulate

Pathogenicity tests were conducted on one-year-old plants of pomegranate var. Bhagwa, raised in earthen pots with dimensions of 45 × 30 cm. The pots were filled with mixed sterile soil and farm yard manure (FYM). Fungal inoculum was prepared in the laboratory by inoculating 5 mm discs cut from the periphery of actively growing culture in a potato dextrose broth and incubated at 26 ± 1 °C for 15 days. Wounds were made at the collar region with a sterilized blade. The wounded area was treated with an insertion of mycelium of *C. Fimbriata* using a sterilized needle and wrapped with a cotton cloth (moistened with sterile distilled water) and plastic film. Then, plants were inoculated by drenching the roots with a spore suspension (10^6^ CFU/mL). The method was replicated thrice, with inoculation of two other plants under polyhouse conditions. The plants inoculated with distilled water served as the control. The plants were regularly watered, and observations were taken every day regarding the development of wilt symptoms. The artificial inoculation with the broth culture of *Ceratocystis Fimbriata* on pomegranate plants was carried out in Plate 1. Once the artificially inoculated plants developed typical symptoms, the disease samples were collected and the organism was re-isolated on a potato dextrose agar medium. Then, it was compared with the original culture to confirm Koch’s postulates and prove pathogenicity.

The isolates collected from different locations were named as shown in Table 1.

### 2.3. Morphological Characterization of the Pathogen

Initially, the pathogen was identified based on the morphological characteristics of the fungus in the culture plate and through microscopic observation of different spores.

Morphological characteristics of *C. fimbriata* were studied in a 15-day-old culture. For that, a mycelial disc diameter in 5 mm was cut from the periphery of the actively growing culture and transferred aseptically to a 90 mm Petri dish containing 20 mL of potato dextrose agar and incubated at 26 ± 1 °C for 15 days. The pathogen was observed for the production of different spores, such as endoconidia, aleurioconidia, perithecia and ascospores. A small quantity of culture was placed on a clean glass slide using a sterilized needle, and the length and breadth of different spores were measured in μm with the help of a fluorescent microscope. Spores were picked randomly for diameter measurement, and three observations were taken for each spore type. The pathogen was also characterized by its colony colour, growth rate and growth pattern on potato dextrose agar.

Molecular characterization was also carried out via ITS (Internal Transcribed Spacer region) sequencing using the ITS region primer sets ITS-1-F (5′-CTT GGT CAT TTA GAG GAA GTA A-3′) and ITS-4-R (5′-TCC TCC GCT TAT TGA TAT GC-3′) to confirm the identity of the pathogen.

### 2.4. Fungal Genomic DNA Extraction

Genomic DNA was extracted from fifteen isolates of *C. fimbriata* using the Cetyl Trimethyl Ammonium Bromide (CTAB) method as described by [5] with slight modifications. All the DNA extracts were diluted using sterilized double-distilled water before PCR amplifications. The DNA was quantified either by gel electrophoresis or nanodrop. The qualitative check for the presence of DNA in samples was confirmed through agarose gel electrophoresis using 0.8% agarose. The purity of DNA was analyzed considering the A260/A280 ratio as determined using Nanodrop (Thermo Scientific, Wilmington, MA, USA).

### 2.5. PCR Amplification of Internal Transcribed Spacer (ITS) Region

Sequences of the ITS region, including the 5.8S gene, were obtained by PCR amplification (DNA sample (50 ng)-1 µL; primers (10 µM)-1 µL; dNTPs(10 mM)-0.5 µL; MgCl_2_(25 mM)-1.5 µL; 10× Taq polymerase buffer −2.5 µL; Taq polymerase (5 U/µL); autoclaved DD water −18.5 µL) from genomic DNA using the primers ITS1 F (5’-CTTGGTCATTTAGAGGAAGTAA-3’) and ITS4 R (5’-TCCTCCGCTTATTGATATGC-3’) following the protocol of [6], with slightly different cycling conditions: initial denaturation at 94 °C for 95 s followed by 35 cycles of denaturation at 94 °C for 35 s, annealing at 52 °C for 60 s and extension at 72 °C for 60 s. The final extension was carried out at 72 ° C for 15 min. The amplification was performed in a 25 μL reaction mixture containing 12.5 µL of 2× PCR master mix, 2 µL each of forward and reverse primers, 2 μL of DNA and finally, 6.5 μL of millipore water. Electrophoresis was carried out for all the products along with 1 kb DNA ladder.

### 2.6. Sequencing of rDNA ITS Region and Sequence Analysis of C. fimbriata Isolates

Sequences of the ITS rDNA region were generated using PCR followed by direct DNA sequencing of the PCR products with primers ITS1 F and ITS4 R

PCR products of six representative *C. fimbriata* isolates from six districts were sent for sequencing. The obtained ITS sequences were analyzed in the NCBI BLAST database. ITS sequences of *C. fimbriata* isolated from pomegranate of various other regions were retrieved from the NCBI database, and all the sequences were analyzed for their similarity/dissimilarity to our *C. fimbriata* isolates. Phylogenetic tree analysis was carried out with Mega X software. The sequences were deposited in GenBank to obtain the accession number.

### 2.7. Genetic Variability Studies of the Pathogenic Isolates

Koch’s postulates were proved on the seedlings for all the obtained isolates. DNA-based molecular markers are essential for the characterization of the genetic relationship of a particular organism. Microsatellite loci, viz., CfAAG8, CfAAG9, CfCCAA9, CfCCAA10, CfCCAA15, CfCAA38, CfCCAA80, CfCAT1, CfCAT3K, CfCAT9X, CfCAT12X, CfCAG5, CfCAG15, CfCAG900, CfGACA60 and CfGACA650, were used to study the genetic variability of the collected isolates of *C. fimbriata* [7].

### 2.8. Band Scoring and Data Analysis

The amplification product was considered as a DNA marker and scored for its presence or absence among the isolates. All reproducible bands were scored manually and analyzed. The data matrix (allelic data) was created, which was further used to construct the dendrogram using the Hierarchical Clustering (UPGMA—Unweighted Pair-Group Method with Arithmetic Averages) method with DARwin 6.0 software.

## 3. Results

### 3.1. Morphological Characterization of the Pathogen

Isolates were found to be diverse in terms of colony colour as they existed in different colours, such as whitish grey, greenish grey, light grey, greyish, grey and dark grey. Among fifteen isolates, six isolates (Cf-3, Cf-5, Cf-8, Cf-11, Cf-12 and Cf-15) exhibited a grey colour, and three were dark grey (Cf-6, Cf-10 and Cf-13), two were light grey (Cf-2 and Cf-9), the Cf-4 and Cf-7 isolates were greyish, isolate Cf-1 was whitish grey and isolate Cf-14 was greenish grey. With respect to colony growth, all the isolates exhibited flat growth and none of them were fluffy. When isolates were observed for colony margin, all the isolates showed a uniform colony margin except isolate Cf-7, which was irregular. Isolates were also assessed for their growth rate, and we found that isolates took 15-35 days to cover entire Petri plates. A total of 15 days was the minimum time required by isolate Cf-5 to cover the whole plate, and this was found to be on par with isolates Cf-15 (17 days), Cf-8 (18 days) and Cf-2 (19 days). In contrast, isolate Cf-14 required the most time to cover the plate (35 days), followed by isolates Cf-1 and Cf-13, which took 30 days to cover the entire Petri plate.

In general, most of the isolates were grey in colour with a flat type of colony growth and uniform/regular margin, and they took 15–24 days to cover the entire Petri plate (Table 2).

Diversity in morphological characteristics such as length and breadth of endoconidia, aleurioconidia, perithecia and ascospores was measured using a fluorescent microscope. All fifteen isolates showed variability with respect to the size of endoconidia, aleurioconidia, perithecia and ascospores.

There were no such differences noticed in the colonization of isolates of *C. fimbriata.* The plants inoculated with pathogen cultures exhibited typical external symptoms, such as yellowing leaves in one or more twigs, followed by drooping and finally complete drying of the entire plant as the disease progressed. Some plants exhibited partial wilting. In such cases, drying and death were observed only in the affected branches. The inoculated plants showed symptoms after 20–25 days of pathogen inoculation and wilted after 28–35 days. Dark greyish to black discolouration was observed in vascular tissues when the affected roots were split open. The pathogen was re-isolated from infected root samples showing typical symptoms, and the pathogen culture was compared with the original isolate and found to be similar, thus proving the pathogenicity.

### 3.2. Molecular Characterization of Ceratocystis fimbriata Using ITS Primers

All fifteen isolates were characterized using ITS gene technology. After 35 cycles of PCR amplification, universal primers (ITS 1 and ITS 4) successfully amplified the entire ITS region and produced an amplicon 600–650 bp in length in all the isolates, indicating that all isolates belong to the genus *Ceratocystis*, confirming the identity of the isolates. After amplification and the subsequent confirmation, the PCR products of six representative isolates from different districts of Southern Karnataka were selected and sent for sequencing at Juniper Life Sciences, Bengaluru, India. The analysis of the obtained sequences was carried out using NCBI, BLAST. Analysis of the ITS gene region revealed the homology of our isolates with other ITS gene sequences in the database. Characterization of six isolates based on ITS gene coding regions revealed that maximum similarity (99.68%) was found with mango isolate (*C. fimbriata*) from China with the accession number KX101050.1. The resemblance of the obtained ITS sequences to the analogues available in the database is presented in Table 3.

Sequences of six *C. fimbriata* isolates of pomegranate were deposited in NCBI GenBank, along with the location of the isolates. The obtained accession numbers are presented in Table 4.

The evolutionary history was developed using the neighbour-joining method. An optimal tree with a sum of branch length of 6.941 was used, and the analysis involved thirty nucleotide sequences. The percentage of replicate trees in which the associated taxa clustered together in the bootstrap test (1000 replicates) is shown next to the branches, and the bootstrap test was conducted with 1000 replications. The evolutionary distances were computed using the p-distance method and are given in the units of the number of base differences per site. All ambiguous positions were removed for each sequence pair (pairwise deletion option). There were a total of 769 positions in the final dataset. Evolutionary analyses were conducted in MEGA X software.

The evolutionary relationship between the isolates of *C. fimbriata* under study with selected GenBank accessions revealed that the isolates under study were put together in the phylogenetic tree with strong branch support and are closely related to the *C. fimbriata* isolate (CF22) from the National Research Centre on Pomegranate (NRCP), Solapur (KU877210.1), and also related to other accessions such as KY580865.1 CF isolate C2866, KY580867.1 CF isolate C2868, KY580869.1 CF isolate C2870, MH793677.1 CF isolate YW3-1 and MH540139.1 CF isolate C3-1GYP (Figure 1).

### 3.3. Genetic Variability Studies of Collected Isolates of Ceratocystis fimbriata

The sixteen microsatellite primers were employed for genetic diversity assessment of *C. fimbriata* isolates collected from various growing regions of Karnataka. Among the 16 microsatellite loci, 12 loci were monomorphic (estimated allele size in bp: CfAAG8 = 160, CfAAG9 = 370, CfCCAA10 = 165, CfCCAA15 = 335, CfCCAA80 = 330, CfCAT1 = 260, CfCAT3K = 310, CfCAT9X = 300, CfCAT12X = 375, CfCAG5 = 315, CfCAG900 = 160 and CfGACA650 = 170). The four polymorphic microsatellite loci had the following respective allele sizes: CfCCAA9 = 170, 210; CfCAA38 = 250, 380, 510; CfCAG15 = 250, 500 and CfGACA60 = 165, 335.

The UPGMA dendrogram (Figure 2) for genetic variation among fifteen isolates of *C. fimbriata* based on alleles of 16 microsatellite loci showed that all fifteen isolates fell into two major clusters. Cluster one consisted of isolate Cf-10 (Thogaragunte Village of Sira Taluk). Cluster two was further divided into two sub-clusters. Sub-cluster one consisted of isolate Cf-2 (Harohalli of Devanahalli Taluk). Sub-cluster two was again divided into five groups. The first group included isolate Cf-13 (Maskal of Hiriyur Taluk), the second group included isolate Cf-14 (Ramajjanahalli of Hosadurga), the third group included isolates Cf-1 (Pura of Devanahalli Village), Cf-4 (Yaluvahalli of Devanahalli Taluk), Cf-6 (GKVK, Bengaluru Urban), Cf-7 (Uyyalappanahalli of Kanakapura Taluk), Cf-8 (Jakkalli of Kollegala Taluk) and Cf-9 (Rangapura of Sira Taluk). The fourth included Cf-5 (Kalenahalli of Bengaluru North) and Cf-11 (Dharmapuri of Hiriyur Taluk). The fifth group comprised Cf-3 (Vijayapura of Devanahalli Taluk), Cf-12 (Javagondanahalli of Hiriyur Taluk) and Cf-15 (Kurubarahalli of Hosadurga).

Interestingly, two (Cf-12 and Cf-15) out of five isolates of the Chitradurga District were found to belong to one sub-cluster (group five). In contrast, the remaining isolates (Cf-11, Cf-13 and Cf-14) fell into other sub-clusters of the same major cluster. Similarly, three (Cf-1, Cf-3 and Cf-4) out of four isolates of Bengaluru Rural district were found to belong to different sub-clusters of the same major cluster. In contrast, isolate Cf-2 was found to belong to another cluster, and isolates from Sira Taluk (Cf-9 and Cf-10) were clustered separately in the dendrogram.

The relatedness among the isolates was evaluated by cluster analysis of the data based on the dissimilarity matrix (Table 5). The dissimilarity coefficient ranged from 0.00 to 0.20 among the isolates. Zero dissimilarity was observed between isolate Cf-1 and Cf-4, Cf-6, Cf-7, Cf-8 and Cf-9; Cf-3 and Cf-12 and Cf-15; and Cf-5 and Cf-11. This indicates that isolates belonged to same cluster/group and were similar to one another, whereas maximum dissimilarity (0.20) was found between isolate Cf-10 and all other isolates, and they were genetically distinct from each other as they were grouped separately in the dendrogram.

## 4. Discussion

### 4.1. Isolation and Identification

The isolation of the fungus *C. fimbriata* was carried out via carrot baiting and tissue isolation techniques using infected roots with the typical symptom of dark greyish-brown streaks. The isolated fungus was confirmed as *C. fimbriata* based on the cultural and morphological characteristics, such as mycelial, endoconidia, aleurioconidia, perithecia and ascospore production, as reported by earlier works [8,9,10,11,12,13]).

### 4.2. Molecular Characterization of Ceratocystis fimbriata Using ITS Primers

In the present investigation, the molecular characterization of fifteen isolates was carried out using ITS gene technology. Amplification of the ITS region of rDNA produced a fragment approximately 600–650 bp in length among all the isolates. It indicated that all belong to the genus *Ceratocystis*, thus confirming the identity of the pathogenic isolates. Characterizing six isolates based on ITS gene coding regions showed that isolate Cf-11 exhibited a maximum similarity of 99.68 percent to mango isolate (*C. fimbriata*) from China, with the accession number KX101050. Similar results with respect to molecular characterization were reported by [14], who confirmed the identity of *C. fimbriata* by the internal transcribed spacer region (ITS 1 and ITS 4), which produced an amplification size of 600–650 bp, and characterization of the fungal isolate (Cf-26) was carried out based on the ITS gene coding region and revealed that maximum similarity (96–99%) was found between *Ceratocystis fimbriata* and *C. manginicans* on pomegranate and mango, respectively.

Similarly, [15] confirmed the identity of twenty isolates of *C. fimbriata* with the internal transcribed spacer region (ITS 1 and ITS 4), which produced an amplification size of 500 bp and confirmed the identity of all the isolates. Molecular identification and characterization of the major fungal pathogen, *C. fimbriata*, was carried out via employing the genus-specific internal transcribed spacer (ITS) universal primer pair ITS1-F/ITS4-R, which was amplified at 600–650 bp [16]. Alam et al. [17] sequenced the rDNA internal transcribed spacer (ITS) region of *C. fimbriata* using ITS1/ITS4 primers, and GenBank BLAST revealed that the ITS sequence (accession no. KX703025) was 99 percent identical to *C. fimbriata*, accession number AM712447, from China.

### 4.3. Genetic Variability Studies of Collected Isolates of Ceratocystis fimbriata

A study on pathogenic variability is essential for breeding disease resistance in the crop improvement program. A potential pathogen often has biodiversity within its population. Variation in the pathogen is a requisite trait for its existence. This variability among the pathogens underlies their diverse nature and ability to survive in the host environment. Variability in pathogens was studied with respect to morphological and molecular characteristics to focus on the variation in *C. fimbriata* isolates collected from different locations during the survey. This serves as a key finding for management aspects.

In the present investigation, the UPGMA dendrogram for genetic variation among fifteen isolates showed that all isolates were grouped into two major clusters as explained earlier, and relationships among the different isolates were evaluated using the cluster analysis of the data based on the dissimilarity matrix.

The findings are in conformity with [15], in which three out of four isolates (NRCP-CF19, NRCP-CF22 and NRCP-CF24) from Maharashtra had fallen into one sub-cluster, and the remaining one, NRCP-CF26, was grouped in the other sub-cluster of the same major cluster, and isolates from different districts of Karnataka did not show any specific pattern of similarity according to geographical region and *C. fimbriata* isolates were found to be distributed randomly throughout two clusters. The result of the BLAST analysis showed that there was no relationship between the isolate and the geographic location. The isolates collected from a particular location were not grouped into the same cluster, and the same trend was observed for all other isolates collected from different places [18]. Santini and Capretti [19] conducted a genetic variability test on the Italian population of *Ceratocystis fimbriata* f. sp. *platani*. Here, a large sample of different isolates was analyzed using Random Amplified Polymorphic DNA (RAPD) and Direct Amplification of Minisatellite-region DNA (DAMD) PCR techniques and compared with several *C.fimbriata* isolates from various hosts growing in different parts of the world. Results indicate that a high level of homogeneity exists in the Italian population of the fungus, whereas certain variability was recognized in isolates from other hosts; diversity among *C. fimbriata* isolates infecting sweet potato have been previously reported to be low [20].

In the present investigation, few isolates from different districts clustered together in the dendrogram and hence showed higher similarity, and few isolates of the same district showed dissimilarity. In general, isolates from other districts did not show a specific pattern of similarity according to geographical region. This may be because wilt fungus is readily introduced to new areas through human intervention through planting material and contaminated tools. It is well-known that the leading cause of fungal spread is pruning operations. This may have influenced not only the spread of the disease but also the genetic variability of the pathogen, especially in areas where it has recently been introduced. All *Ceratocystis* species can produce perithecia and ascospores through selfing via unidirectional mating-type switching [21]. Thus, even natural populations would be expected to have relatively low genetic diversity. [22], and introduced populations are expected to be essentially clonal and have minimal genetic diversity [23]

## 5. Conclusions

All the fifteen isolates characterized using ITS gene technology produced an amplification size of 600–650 bp and were found to belong to the genus *Ceratocystis*. The UPGMA dendrogram for genetic variation among the isolates of *C. fimbriata* showed that all the isolates fell into two major clusters, and the isolates from different districts did not show a specific pattern of similarity according to geographical region. The dissimilarity coefficient ranged from 0.00 to 0.20 among the pathogenic isolates. Isolates Cf-1, Cf-3, Cf-4, Cf-5 Cf-6, Cf-7, Cf-8, Cf-9, Cf-11, Cf-12 and Cf-15 were found to be highly similar to each other as their dissimilarity coefficient was zero, and maximum dissimilarity (0.20) was found between isolate Cf-10 and all other isolates, which were seen to be genetically distinct from each other.

## Figures and Tables

**Figure 1 jof-08-01276-f001:**
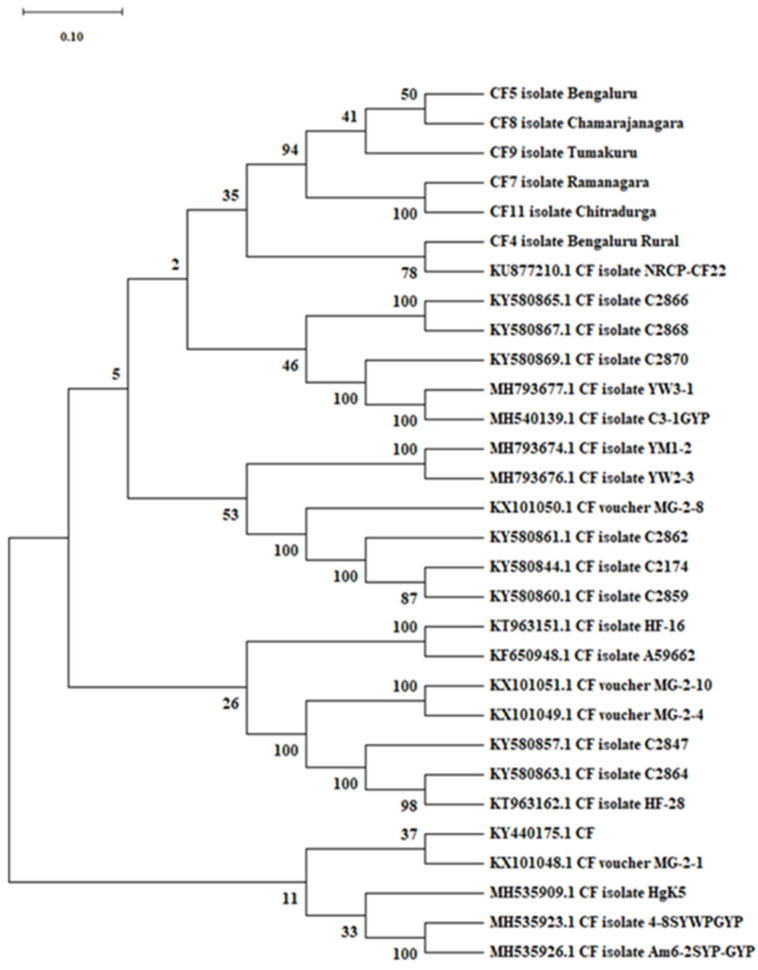
Phylogenetic tree showing evolutionary relationships between isolates of *Ceratocystis fimbriata* and selected GenBank accessions.

**Figure 2 jof-08-01276-f002:**
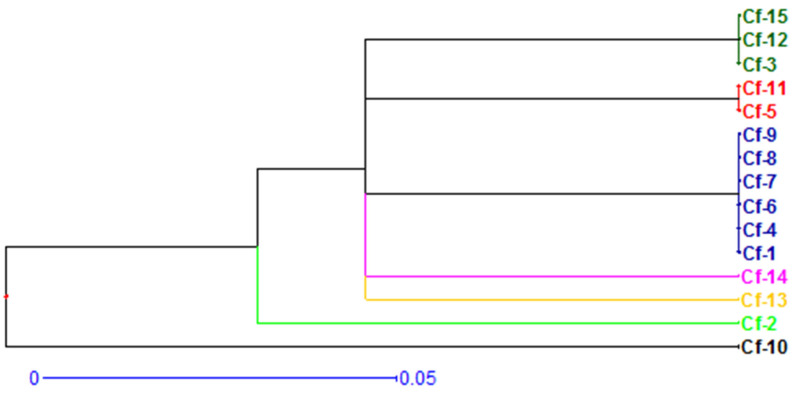
UPGMA dendrogram representing genetic variation among fifteen isolates of *Ceratocystis fimbriata* based on alleles of sixteen microsatellite loci (SSR markers).

**Table 1 jof-08-01276-t001:** Isolates of *Ceratocystis fimbriata* collected from different southern districts of Karnataka used in this study.

Sl. No.	Isolate	Name of the Place		
		Village	Taluk	District
1	Cf-1	Pura	Devanahalli	Rural Bengaluru
2	Cf-2	Harohalli		
3	Cf-3	Vijayapura		
4	Cf-4	Yaluvahalli		
5	Cf-5	Kalenahalli	North Bengaluru	Bengaluru
6	Cf-6	GKVK, Hebbal		
7	Cf-7	Uyyalappanahalli	Kanakapura	Ramanagara
8	Cf-8	Jakkalli	Kollegala	Chamarajanagara
9	Cf-9	Rangapura	Sira	Tumakuru
10	Cf-10	Thogaragunte		
11	Cf-11	Dharmapura	Hiriyur	Chitradurga
12	Cf-12	Javagondanahalli		
13	Cf-13	Maskal		
14	Cf-14	Ramajjanahalli	Hosadurga	
15	Cf-15	Kurubarahalli		

**Table 2 jof-08-01276-t002:** Cultural characteristics of different isolates of *Ceratocystis fimbriata* on potato dextrose agar.

Sl. No.	Isolate	Colony Colour	Type of Colony Growth	Type of Margin	Growth Rate (cm)				Days Taken for Complete Growth
					Growth at 10 DAIs	Growth at 15 DAIs	Growth at 20 DAIs	Growth at 25 DAIs	
1	Cf-1	Whitish grey	Flat	Uniform	3.63	6.13	7.13	7.83	30
2	Cf-2	Light grey	Flat	Uniform	5.76	8.60	9.00	9.00	19
3	Cf-3	Grey	Flat	Uniform	4.06	6.67	9.00	9.00	20
4	Cf-4	Greyish	Flat	Uniform	5.23	8.30	8.73	9.00	23
5	Cf-5	Grey	Flat	Uniform	6.80	9.00	9.00	9.00	15
6	Cf-6	Dark grey	Flat	Uniform	4.33	7.26	8.32	9.00	24
7	Cf-7	Greyish	Flat	Irregular	4.53	7.36	7.98	9.00	26
8	Cf-8	Grey	Flat	Uniform	4.93	8.23	9.00	9.00	18
9	Cf-9	Light grey	Flat	Uniform	6.20	8.83	8.92	9.00	23
10	Cf-10	Dark grey	Flat	Uniform	4.20	7.50	9.00	9.00	19
11	Cf-11	Grey	Flat	Uniform	5.20	7.00	8.60	9.00	22
12	Cf-12	Grey	Flat	Uniform	4.86	7.13	8.86	9.00	21
13	Cf-13	Dark grey	Flat	Uniform	2.63	3.83	5.33	6.46	30
14	Cf-14	Greenish grey	Flat	Uniform	4.30	5.26	5.91	6.56	35
15	Cf-15	Grey	Flat	Uniform	5.96	8.80	9.00	9.00	17
								S.Em ±	1.64
								CD @ 1%	4.81

DAIs: days after inoculation.

**Table 3 jof-08-01276-t003:** Nucleotide sequence similarity of isolates of *Ceratocystis fimbriata* infecting pomegranate to selected GenBank accessions.

Sl. No.	Isolate	Closest Match	Country	Query Cover (%)	Percent Similarity (%)	Query Length (bp)	Accession No.
1	Cf-4	*C. fimbriata* isolate C2174	China	99	98.43	656	KY580844.1
		*C. fimbriata*	China	99	98.42	675	KY440175.1
		*C. fimbriata* HgK5	China	98	98.88	769	MH535909.1
		*C. fimbriata* isolate 4-8SYWPGYP	China	97	99.03	759	MH535923.1
		*C. fimbriata* isolate C2870	China	99	98.42	648	KY580869.1
2	Cf-5	*C. fimbriata* voucher MG-2-10	China	96	90.10	660	KX101051.1
		*C. fimbriata* voucher MG-2-8	China	96	90.10	640	KX101050.1
		*C. fimbriata* voucher MG-2-4	China	96	90.10	651	KX101049.1
		*C. fimbriata* voucher MG-2-1	China	96	90.10	652	KX101048.1
		*C. fimbriata* YM1-2	China	96	89.95	635	MH793674.1
3	Cf-7	*C. fimbriata* isolate C2866	China	98	98.42	656	KY580865.1
		*C. fimbriata* isolate C2864	China	98	98.43	665	KY580863.1
		*C. fimbriata* isolate C2847	China	98	98.43	666	KY580857.1
		*C. fimbriata* isolate C2862	China	98	98.42	653	KY580861.1
		*C. fimbriata* isolate HF-28	China	98	98.43	665	KT963162.1
4	Cf-8	*C. fimbriata* isolate AM6-2SYP-GYP	China	96	99.52	759	MH535926.1
		*C. fimbriata* isolate YW2-3	China	96	99.52	635	MH793676.1
		*C. fimbriata* isolate YW3-1	China	95	99.51	629	MH793677.1
		*C. fimbriata* isolate C3-1GYP	China	95	99.51	627	MH540139.1
		*C. fimbriata* isolate NRCP-CF22	Bagalkote, India	97	98.72	623	KU877210.1
		*C. fimbriata* isolate NRCP-CF24	Bagalkote, India	96	99.03	623	KU877211.1
5	Cf-9	*C. fimbriata* isolate C2174	China	99	99.06	656	KY580844.1
		*C. fimbriata* isolate C2864	China	98	99.06	665	KY580863.1
		*C. fimbriata* isolate C2847	China	98	99.06	666	KY580857.1
		*C. fimbriata* isolate C2866	China	99	98.90	656	KY580865.1
		*C. fimbriata* isolate C2864	China	99	98.90	665	KY580863.1
6	Cf-11	*C. fimbriata* voucher MG-2-10	China	97	99.52	660	KX101051.1
		*C. fimbriata* voucher MG-2-8	China	97	99.68	640	KX101050.1
		*C. fimbriata* voucher MG-2-4	China	97	99.52	651	KX101049.1
		*C. fimbriata* voucher MG-2-1	China	97	99.52	652	KX101048.1
		*C. fimbriata* isolate C2868	China	97	99.36	653	KY580867.1

**Table 4 jof-08-01276-t004:** Accession numbers of *Ceratocystis fimbriata* nucleotide sequences submitted to NCBI, GenBank.

Sl. No.	Isolates	Districts	Accession Numbers
1	Cf-4 (ITS 1)	Bengaluru Rural	MZ749649
	Cf-4 (ITS 4)		MZ749650
2	Cf-5 (ITS 1)	Bengaluru Urban	MZ749663
	Cf-5 (ITS 4)		MZ749664
3	Cf-7 (ITS 1)	Ramanagara	MZ749665
	Cf-7 (ITS 4)		MZ749666
4	Cf-8 (ITS 1)	Chamarajanagara	MZ749671
	Cf-8 (ITS 4)		MZ749672
5	Cf-9 (ITS 1)	Tumakuru	MZ749741
	Cf-9 (ITS 4)		MZ749742
6	Cf-11 (ITS 1)	Chitradurga	MZ749730
	Cf-11 (ITS 4)		MZ749731

**Table 5 jof-08-01276-t005:** Dissimilarity matrix of fifteen isolates of *Ceratocystis fimbriata*.

	Cf-1	Cf-2	Cf-3	Cf-4	Cf-5	Cf-6	Cf-7	Cf-8	Cf-9	Cf-10	Cf-11	Cf-12	Cf-13	Cf-14	Cf-15
**Cf-1**	1.00														
**Cf-2**	0.13	1.00													
**Cf-3**	0.10	0.13	1.00												
**Cf-4**	0.00	0.13	0.10	1.00											
**Cf-5**	0.10	0.13	0.10	0.10	1.00										
**Cf-6**	0.00	0.13	0.10	0.00	0.10	1.00									
**Cf-7**	0.00	0.13	0.10	0.00	0.10	0.00	1.00								
**Cf-8**	0.00	0.13	0.10	0.00	0.10	0.00	0.00	1.00							
**Cf-9**	0.00	0.13	0.10	0.00	0.10	0.00	0.00	0.00	1.00						
**Cf-10**	0.20	0.20	0.20	0.20	0.20	0.20	0.20	0.20	0.20	1.00					
**Cf-11**	0.10	0.13	0.10	0.10	0.00	0.10	0.10	0.10	0.10	0.20	1.00				
**Cf-12**	0.10	0.13	0.00	0.10	0.10	0.10	0.10	0.10	0.10	0.20	0.10	1.00			
**Cf-13**	0.10	0.13	0.10	0.10	0.10	0.10	0.10	0.10	0.10	0.20	0.10	0.10	1.00		
**Cf-14**	0.10	0.13	0.10	0.10	0.10	0.10	0.10	0.10	0.10	0.20	0.10	0.10	0.10	1.00	
**Cf-15**	0.10	0.13	0.00	0.10	0.10	0.10	0.10	0.10	0.10	0.20	0.10	0.00	0.10	0.10	1.00

## Data Availability

Not applicable.

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
