# Peer review of "Molecular Characterization and Genetic Variation in Ceratocystis fimbriata Ell. and Halst. on Pomegranate"

_jof, 2022, doi:10.3390/jof8121276_

Round 1

Reviewer 1 Report

The article is important from the point of view of basic knowledge, however, I believe that the authors can include in the introduction and discussion how their findings will contribute to the implementation of disease management strategies. In my opinion, this would broaden the impact and interest of the manuscript. In addition, the authors must write the perspectives of the investigation based on the results obtained.

Additionally, in lines 15, 42, 64, 103, 105, 135, 138, 150, 152, 165, 173, 2015, 2017, 222, 225, 228, 229, 237, 287 C. fimbriata should be written in italics.

In line 115 change Ceratocystis fimbriata for C. fimbriata

Figure 1 should improve its quality since it cannot be observed properly.

Author Response

Detailed response to reviewers comments is attached as a separate file

Reviewer 2 Report

Dear Authors,

Review of “Molecular characterization and genetic variation of Ceratocystis fimbriata Ell. and Halst. on pomegranate”

Introduction

lines 42 – 45 and 46-48 contain the same information,

Please write more information about the cultivation area of pomegranate plants, the biology of C. fimbriata, its isolation from plant tissues. Are there problems with isolating C. fimbriata from plants? C. fimbriata is soilborne ascomyce so whether the soil was tested for the presence fungus.

Materials and methods

2.1

please indicate the year or years of sampling,

line 53 The infected samples of pomegranate were collected…, please write from where (from the infected stem, root and branch tissues), What were the symptoms of pomegranate infection in orchards?

How many samples were taken? Was a C. fimbriata isolate obtained from each sample?

It needs to be completed

lines 55-56 isolations were done following the standard method, please write reference and briefly describe

2.2

Not to describe the whole method of DNA isolation but only modifications.

2.3

please write the concentrations of reagents reaction amplification

2.4

Lines 101-102 Remove primer sequences, they are above

2.5

Line 110 Genetic variability studies of the pathogenic isolates

Whether Koch's postulates were done?

Whether pathogenicity tests were done on  seedlings or part of plant tissues (leaves/stem) by obtained  isolates ?

Results

Lines 125-129 All the fifteen isolates were characterized using ITS gene technology and after 35 cycles of PCR amplification, universal primers (ITS 1 and ITS 4) were successfully amplified the entire ITS region and produced an amplicon of 600-650 bp length in all the isolates and indicated that, all isolates belongs to the genus Ceratocystis confirming the identity of the isolates.

Has the morphological identification of the isolates been done?

Please describe this aspect in more detail in both Material and Methods and the results.

The discussion contains many sentences from the results and could be extended to include more literature.

Author Response

Detailed response to reviewers comments is attached

Round 2

Reviewer 2 Report

Dear Authors,

Review of “Molecular characterization and genetic variation of Ceratocystis fimbriata Ell. and Halst. on pomegranate”

Comments for Authors

2.4

please write the concentrations of reagents reaction amplification

2.5

Remove primer sequences, they are above

2.6 Genetic variability studies of the pathogenic isolates

Sentence “Koch’s postulates were proved on the seedlings for all the obtained isolates”

should be in a subsection 2.1. Please describe briefly how it was made.

In the Results section, describe the results of colonization seedlings pomegranate by obtained isolates. Were there differences in colonization plant tissues between isolates of C. fimbriata.

Author Response

Dear Reviewer

We have added all the corrections suggested by you. We profoundly thank you for your suggestions towards improvement of the paper.
